# How the Supramolecular Nature of Lignohumate Affects Its Diffusion in Agarose Hydrogel

**DOI:** 10.3390/molecules25245831

**Published:** 2020-12-10

**Authors:** Martina Klučáková, Michal Kalina, Vojtěch Enev

**Affiliations:** Materials Research Centre, Faculty of Chemistry, Brno University of Technology, Purkyňova 118/464, 612 00 Brno, Czech Republic; kalina-m@fch.vutbr.cz (M.K.); enev@fch.vutbr.cz (V.E.)

**Keywords:** lignohumate, supramolecular nature, diffusion, spreading, hydrogel

## Abstract

Lignohumate, as an industrially produced analog of natural humic substances, is studied from the point of view of its diffusion properties. This work focuses on its permeation ability, important in agricultural and horticultural applications, connected with its penetration into plant organs as leaves and roots. The hydrogel based on agarose was used as a model material for the diffusion of lignohumate. Two types of experiments were realized: the diffusion of lignohumate in the hydrogel diffusion couple and the diffusion of lignohumate from its solution into hydrogel. The diffusion coefficient of lignohumate in the hydrogel was determined and used for the modelling of the time development of concentration profiles. It was found that the model agrees with experimental data for short times but an accumulation of lignohumate in front of the interface between donor and acceptor hydrogels was observed after several days. The particle size distribution of lignohumate and changes in the *E*_4_/*E*_6_ ratio used as an indicator of molecular weight of humic substances were determined. The results showed that the supramolecular structure of lignohumate can react sensitively to actual changes in its environs and thus affect their mobility and permeability into different materials. A filtration effect at the interface can be observed as an accompanying phenomenon of the re-arrangement in the lignohumate secondary structure.

## 1. Introduction

Lignohumate is an industrially produced analog of natural humic substances, produced by thermal processing of technical lignosulfonate, which is based on the oxidation and hydrolytic destruction of lignin-containing raw material [1,2]. It can promote plant growth and therefore it is widely used for agricultural and horticultural purposes [2,3,4,5,6,7,8]. It was found that its application can reduce leaching of nitrogen into the soil solution [2], reduce the toxic effect of pesticides, improve the nitrogen balance of soils, increase the basal respiration of the soil, and has a positive effect on soil biological activity [9].

Lignohumate together with other humic and humic-like substances has a complex conformational arrangement and can be perceived as supramolecular structures, the functioning of which is determined by the composition, size of molecular units, and weak intermolecular forces [10]. A major aspect of the supramolecular arrangement is its stabilization by weak dispersion forces, where hydrophobic and hydrogen bonds are responsible for their apparent large molecular size [11]. In solutions, humic and humic-like substances can be micelle-like, supra-molecular assemblies of small entities and their secondary structure can change significantly with changing concentration and pH [12,13,14,15,16,17,18,19,20,21]. In soils, humic substances are spatially arranged in decreasing order of polarity. This means that highly polar supramolecular subunits shield less polar subunits against the free soil solution and form layers of decreasing polarity [22].

Initially, the term “lignohumate” was characterized by Springer [23] as the humic material with lower ripeness and nitrogen content and a relatively high proportion of sulfonic acidic groups (as distinguished from the “true” soil humic material). Lignohumate has structure and properties similar to humic substances isolated from native matrices. During its production, the lignosulfonate is transformed into a humic-like material (by means of a higher temperature and pressure). Chemical changes ongoing in the transformation are similar to the humification of plant litter in nature [2]. The final product (lignohumate) usually has a lower nitrogen content and higher content of sulfonic acidic groups as well as other *O*-containing groups (hydroxyl, carbonyl, carboxyl, ether and ester ones) in comparison with humic substances isolated from peat, soils, and sediments. In comparison with native humic substances, the lignohumate can be characterized by the presence of simple structural components of wide molecular heterogeneity, smaller molecular size and weight [2,24].

Present studies of lignohumate are focused mainly on its agricultural and horticultural applications [2,3,4,5,6,7,8]; therefore, studies dealing with its structural characterization and other properties are relatively scarce. Some works [2,24] used chemical (elemental analysis) and spectral (UV/VIS, FT-IR nad ^13^C-NMR) methods for the characterization of lignohumate and its comparison with humic material extracted from different natural matrices. The diffusion of lignohumate through plant cuticles was studied in our previous work [8]. The cuticle was placed between two cuvettes filled by agarose hydrogels. The donor hydrogel was enriched by lignohumate, the acceptor hydrogels was prepared without it (i.e., with zero initial concentration of lignohumate), and the transport of lignohumate through cuticle was monitored. This work is focused on the intrinsic diffusion in the hydrogels. The diffusion couple was assembled by means of the donor and acceptor hydrogels, but no cuticle was placed between them and both hydrogels were in immediate contact at the interface. In consideration of the heterogeneous structure of lignohumate [1,2,24] and unusual results obtained for transport properties of agarose hydrogels containing humic substances [25], the diffusion of lignohumate in agarose hydrogels should be investigated in detail. If the diffusion is realized in the two hydrogels, the lignohumate has a concentration gradient in both parts (donor and acceptor). In contrast, the solution can be stirred, therefore it has no concentration gradient in the donor solution. Therefore, the diffusion of lignohumate from its aqueous solution into agarose hydrogel was also investigated in this work.

## 2. Results and Discussion

The first experimental task was to determine the diffusion coefficient of lignohumate in the agarose hydrogel on the basis of experimental data obtained for the diffusion couple described above. If the couple is composed of hydrogels of the same nature, differing only in the lignohumate content, the diffusion coefficient should be the same in both parts of couple—the donor and acceptor hydrogels—in conditions in which the diffusion coefficient in not dependent on the concentration of diffusing particles [26,27,28]. Mathematical description of diffusion in the diffusion couple is relatively easy. Before the start of diffusion, the donor hydrogel has a constant concentration of lignohumate *c*_0_ in the whole cuvette volume, whereas its concentration in the acceptor hydrogel is zero. If both parts of the diffusion couple can be considered as semi-infinite mediums (meaning that the concentrations on the outside borders of the couple are constant and do not change in time during diffusion), the solution of second Fick law is [26,27,28]
(1)$$c_{x,t}=\frac{1}{2}c_{0}erfc\frac{x}{2\sqrt{D_{h}t}}\;,$$
where *c_x,t_* is the concentration of lignohumate at position *x* in time *t* and *D_h_* is the effective diffusion coefficient of lignohumate in hydrogel. It can be seen that the concentration of the diffused component on the interface (*c_i_*) is time independent and equal to *c*_0_/2.

Total diffusion flux *m_t_* which goes through the interface between donor and acceptor hydrogels (*x* = 0) in time *t* can be calculated as [26,27,28]
(2)$$m_{t}=c_{0}\sqrt{\frac{D_{h}t}{\pi }}\;.$$

Equation (2) assumes that there is no accumulation of lignohumate at the interface, i.e., the diffusion flux from the donor hydrogel to the interface is equal to the diffusion flux from the interface into the acceptor hydrogel.

In Figure 1, the examples of experimental data are shown. As expected, the amount of lignohumate in the donor hydrogel decreased gradually as it diffused through the interface into the acceptor hydrogel. Our results confirm that the experimentally determined dependency of *m_t_* on $\sqrt{t}$ is linear up to a critical time, when the requirement of semi-infinite mediums is not fulfilled. Therefore, only the data corresponding with the abovementioned conditions are fitted by Equation (2).

The diffusion coefficient of lignohumate in agarose hydrogel *D_h_* was calculated on the basis of Figure 1b and Equation (2). Its value was determined as 1.46 × 10^−10^ m^2^/s (±8.66 × 10^−12^ m^2^/s), which agrees with values determined for humic substances by some other authors [29,30,31,32,33]. Cornel et al. [29] studied the diffusion of humic acids fractionated by means of ultrafiltration. They determined the diffusion coefficients between 3.8 × 10^−11^ and 1.6 × 10^−10^ m^2^/s dependent on pH, ionic strength, temperature, and humic molecular weight. The dependence of the diffusion coefficient on molecular weights (according to the Stokes-Einstein equation) was weaker in comparison with the results obtained for synthetic polymers. This anomalous behavior of humic acids was interpreted to be an artefact of the determination of molecular weights by ultrafiltration. The authors concluded that true molecular weights of humic fractions were substantially lower than the nominal values of ultrafiltration. Pinheiro et al. [30] used voltammetric techniques for the determination of diffusion coefficients of humic and fulvic acids and their metal complexes. They obtained results in the range of from 5 × 10^−12^ to 2 × 10^−11^ m^2^/s for humic acids and their complexes and from 6 × 10^−11^ to 1.2 × 10^−12^ m^2^/s for fulvic acids and their complexes. Lead et al. [31] studied the diffusion of humic and fulvic acids isolated from the Suwannee river (standard of International Humic Substances Society) by means of the fluorescence correlation spectroscopy. They determined the diffusion coefficients in the range of (2–3) × 10^−10^ m^2^/s. They observed the aggregation of humic substances at lower pH values. No effect of concentration and ionic strength was observed. Lead and Wilkinson [32] used the same method for the determination of diffusion coefficients of different humic substances in water and agarose hydrogel. In general, their results were in the range of (1.9–2.3) × 10^−10^ m^2^/s for all samples under all conditions. Diffusion coefficients determined for hydrogel were approximately 10–20% lower in comparison with water, mainly due to increased path lengths and tortuosity. They assumed that large aggregates of humic substances were excluded from the hydrogel. In their subsequent work [33], the results obtained by means of the fluorescence correlation spectroscopy and classical diffusion cells were compared. Decreases in diffusion coefficients observed for some humic samples in hydrogel were too large to be explained by the tortuosity and obstructive effects of the hydrogel pore structure. Very hydrophobic humic substances were probably prevented from penetrating the hydrogel because of their aggregation. They stated that a potential complexation of humic substances with the hydrogel fibers is not important and can be neglected. In contrast, the diffusion coefficients of humic acids in water determined in ref. [34] were much lower. They varied between 2.3 × 10^−12^ and 8.6 × 10^−12^ m^2^/s in the case of values measured by means of dynamic light scattering. The values based on voltammetry were in the range of 4.4 × 10^−12^ and 1.3 × 10^−11^ m^2^/s.

The value of the diffusion coefficient (determined in this work) was used for the calculation of the diffusion profile according to Equation (1). A comparison of experimental data with the computed curve is shown in Figure 2. It seems that our experimental data are in a good agreement with the simple mathematical model and the concentration of lignohumate at the interface between donor and acceptor hydrogels (*c_i_*) can be considered as the half *c*_0_. However, an interesting phenomenon was observed when the diffusion was continued for several days. We can see that that one point close to the interface from the donor part is higher than the half *c*_0_ after one day of diffusion (Figure 2), as well as when the diffusion was prolonged (Figure 3a). After several days, the concentration of lignohumate on the outside border of the donor part decreased and the concentration close to the interface increased (Figure 3b). A concentration jump was observed at the interface for longer times. What is the reason for this? It is not easy to explain. One possibility is a filtration effect. The pore size of agarose hydrogel was determined by a simple spectrophotometric method [35] in our previous work [36]. Their average diameter was determined as 360 nm. The Z-average diameter of lignohumate decreased with the increasing concentration of its solution and its value was ~37 nm for 0.1% wt. content (Figure 4a). This means that the pore size exceeds the Stokes hydrodynamic radius of lignohumate by almost ten times.

The polydispersity for 0.1% wt. lignohumate content was approximately 0.5, zeta potential around −40 mV. More negative zeta potential indicated a higher stability of particles of lignohumate. Many authors (e.g., [37,38,39]) used the ratio of absorbance at 465 to 665 nm (*E*_4_/*E*_6_, so called index of humification) as an indicator of the average molecular weight and size and with the oxygen content of humic materials. The *E*_4_/*E*_6_ ratio is usually <5 for humic acids [1,24,37,38,40]. However, it was found that the values of the humification index *E*_4_/*E*_6_ is higher than 5 for the content of lignohumate above 0.01% wt. (see Figure 4a). It may be indicative of the presence of *O*-containing functional groups (hydroxyl, carbonyl, carboxyl, and ester groups) and lower molecular weight.

As can be seen, the *E*_4_/*E*_6_ ratio corresponds with the Z-average size of lignohumate, and therefore it may be considered as an indicator of its molecular size during the diffusion in the agarose hydrogel. However, the situation is more complex. In Figure 5, the intensity, volume, and number particle size distributions of lignohumate are shown. While the intensity and volume size distributions were tri-modal, their number-based distribution was mono-modal because the number of big particles occupying a large volume was very low. Despite the fact that the majority of lignohumate particles are smaller than the average pore size in the hydrogel, some of them can be limited in their motion, which can result in their higher concentration close to the interface. Some studies showed that the average size diameter is not a suitable parameter for characterizing humic substances and should be used only conditionally as the apparent mean particle size in comparison with other works [41,42]. As can be seen in Figure 4 and Figure 5, the relatively high polydispersity and multi-modal character of particle size distributions can support this conclusion. This means that the lignohumate can contain more fractions with different mobilities. The knowledge of particle size distribution is thus very important. The colloidal stability can thus affect their molecular organization and re-arrangement in the studied system.

In the case of globular particles, their diffusion coefficient decreases with the square of diameter:*D* = *k**_B_**T*/(*6**π**η**r*).(3)
where *D* is the diffusion coefficient (equal to *D_h_* in hydrogel and to *D_s_* in solution), *k_B_* is the Boltzmann constant, *T* is the temperature, *η* is the viscosity of the medium and *r* is the diameter of the diffusing particle [26,27,36]. The re-arrangement (e.g., aggregation and destruction of weak bonds in supramolecular structure of lignohumate) can result in changes in the size of particles, their mobility and the potential accessibility of active sites for other constituents in present natural systems.

As can be seen in Figure 6, the *E*_4_/*E*_6_ ratio of lignohumate in the hydrogel changes with the position in the diffusion couple. The values of absorbance in the acceptor hydrogel for distances >20 mm were too low, and therefore the *E*_4_/*E*_6_ ratio was not calculated for these distances from the interface. If we compare Figure 3 and Figure 6, we can see that the increase in lignohumate concentration in the donor hydrogel is connected with the decrease in the *E*_4_/*E*_6_ ratio. This means that bigger particles cumulated in front of the interface and can pass through it only with difficulty. Simultaneously, the donor hydrogel is deprived of smaller particles diffused through the interface into the acceptor hydrogel, which caused the increase in the *E*_4_/*E*_6_ ratio past the interface. Changes observed in particle size distributions during the diffusion can result in changes in lignohumate molecular organization. Smaller particles are leaving the donor hydrogel and the bigger particles accumulate in front of the interface. If smaller particles were constituents of the supramolecular structure of lignohumate and disappeared from the donor hydrogel, the remaining bigger particles accumulated in a relatively thin layer re-arranged and a new spatial organization of present particles can be established. New bonds between particles can form and others are destroyed. The supramolecular structure changes are dependent on the amount and composition of the present particles.

In order to investigate the diffusion of lignohumate in agarose hydrogel in detail and understand the observed differences between the mathematical model for the diffusion couple (Equation (1)) and the obtained experimental data (Figure 3), the diffusion of lignohumate from its solution into the hydrogel was studied. As described above, the particle size distribution and the stability of molecular organization are strongly influenced by the content of lignohumate in its solution. The decrease in its concentration is connected with the decrease in the colloidal stability and increase in the average diameter and polydispersity. On the other hand, the decrease in lignohumate content in the solution caused by its diffusion into the hydrogel is connected with the depletion of smaller particles and re-arrangement of the supramolecular structure of lignohumate. As we can see in Figure 7, the *E*_4_/*E*_6_ ratio decreases with the continuing diffusion, which indicates the increase in the particle diameter. The increase in particle size is connected with a faster mobility of smaller particles and their diffusion into the hydrogel as well as the re-organization of the supramolecular structure of lignohumate. In contrast, the *E*_4_/*E*_6_ ratio in the vicinity of the interface in hydrogel has a maximum. This means that the first layer in the hydrogel is gradually saturated by smaller particles from the solution and these particles can leave it and diffuse into more distant layers of the hydrogel. It seems that particles of lignohumate in the hydrogel are bigger than its particles dissolved in solution (mainly at the start of the experiment). It is impossible to determine a particular size on the basis of the value of *E*_4_/*E*_6_ ratio. This ratio should be considered as an indicator of the changes in particle size, and therefore it can indicate such processes as the formation of aggregates and disintegration of bigger agglomerates. It is necessary to take into consideration many factors and circumstances. One of these is the fact that it is difficult to compare particle size in different mediums—in solution and in hydrogel. The absorbance (used for the calculation of *E*_4_/*E*_6_ ratio) must be corrected according to the effect that the absorbance of pure medium has subtracted from that measured for lignohumate in a given medium at a given wavelength. Our previous results [14,25] show that the addition of humic substances into agarose hydrogels can affect their behavior mainly from the point of view of its rheological properties. The hydrogels enriched by humic substances were more liquid in comparison with the pure agarose hydrogel and had a lower ability to resist mechanical stresses. Therefore, the background correction by means of the absorbance of pure agarose hydrogel can shift the values of *E*_4_/*E*_6_ ratio at the interface between different mediums.

On the other hand, the behavior of lignohumate in different mediums can differ. Although the size of pores is sufficiently large in the comparison with lignohumate particles [36], the environs of lignohumate can influence its molecular organization. The secondary structure of lignohumate is thus sensitive to its concentration and the medium in which it is located. The particle size distribution and the *E*_4_/*E*_6_ ratio can change with the position in hydrogel as a result of proceeding diffusion, different mobilities of particles, their content in a given locality, etc. Therefore, the supramolecular structure of lignohumate can react sensitively on actual changes in the lignohumate arrangement and surrounding medium, which can affect their mobility and permeability into different materials. As mentioned above, the lignohumate is frequently used in agricultural and horticultural applications. Nowadays, foliar applications of fertilizers and bio-stimulants complement standard root fertilization [8,43]. Diffusion processes play an important role in both methods of fertilization because the application of fertilizers is connected with the penetration into plant organs such as leaves and roots. The supramolecular structure of lignohumate and its ability regarding structural re-arrangement thus can influence its mobility and permeability into plants.

## 3. Materials and Methods

### 3.1. Chemicals

Agarose (AG; routine use class; CAS 9012-36-6) was purchased from Sigma-Aldrich (St. Luis, MO, USA). Lignohumate A was kindly provided by Amagro (Prague, Czech Republic). It represents a commercial mixture of humates and fulvates prepared by thermal conversion of technical lignosulfonates under strictly controlled conditions [1,2,44]. Its main characteristics, such as elemental composition and structural features, can be found in refs. [1,8,24].

The particle size distribution and zeta potential of lignohumate in water (0.005–10 g·dm^−3^) were measured by means of a Zetasizer Nano ZS with backscattering detection (Malvern Instruments Ltd., Worcestershire, UK) The diameter of particles was calculated by means of the Z-average function (based on the Stokes-Einstein equation, assuming particles to be spherical) [29,40,45,46].

### 3.2. Preparation of Hydrogels

The method of preparation of hydrogels was described in detail in our previous studies [8,14,25,36]. The preparation of hydrogels was based on the thermo-reversible gelation of AG aqueous solution. AG was dissolved in deionized water or in an aqueous solution of lignohumate, then heated (80 °C) and stirred to obtain a transparent solution, and finally sonicated to remove gases. Afterwards, the solution was poured into the PMMA spectrophotometric cuvette (inner dimensions: 10 × 10 × 45 mm). The donor hydrogel for diffusion couple was prepared from 1% wt. AG solution containing 0.1% wt. of lignohumate. The acceptor hydrogel was prepared using 1% wt. AG solution (free of lignohumate). The pure agarose hydrogel prepared in the same way as the acceptor one was also used for the diffusion experiments monitoring the transport of lignohumate from its aqueous solution (0.1% wt.).

The size of pores in agarose hydrogel was determined by the spectrophotometric method published by Aymard et al. [35]. The average diameter of pores was determined as 0.36 μm [36].

### 3.3. Diffusion Experiments

Two types of diffusion experiments were realized. The first one was the diffusion of lignohumate in the diffusion couple formed by the donor and acceptor hydrogels, while the second one was the diffusion of lignohumate from its solution into the hydrogel.

#### 3.3.1. Diffusion Couple

The diffusion couple was realized by connecting two cuvettes filled by agarose hydrogels, as described in ref. [8]. The donor hydrogel contained homogeneously dispersed lignohumate (0.1% wt.), and the acceptor hydrogel was prepared as a pure one without lignohumate. During the diffusion experiments, all the diffusion couples were placed in a closed container above water level (in order to maintain constant humidity of the surroundings). Experimental conditions (relative humidity 100%, temperature 25 °C) were held constant during the whole experimental period. The diffusion from the donor into the acceptor part of couple was monitored in time. The concentration profiles of lignohumate in both hydrogels were determined by means of UV/VIS spectrometry (Varian Cary 50 UV/VIS spectrophotometer, Agilent Technologies Inc., Santa Clara, CA, USA). The spectra were measured at various distances from the interface by means of the spectrophotometer equipped with the special accessory providing controlled fine vertical movement of the cuvette in the spectrophotometer [8,25,47]. The collected spectra (at different positions in hydrogels) were used for the calculation of the concentration profiles of lignohumate in the diffusion couple. The diffusion fluxes were determined as the amount of lignohumate transported from the donor hydrogel into the acceptor hydrogel through the interface (per unit of area). These diffusion experiments were performed with ten repetitions and data are presented as average values with standard deviation bars.

#### 3.3.2. Diffusion from Solution into Hydrogel

The diffusion from the lignohumate solutions into the pure agarose hydrogel was realized by the following experimental arrangement. Ten cuvettes with agarose hydrogels (without lignohumate) were placed into five vessels filled by the solution of lignohumate (two cuvettes in 100 cm^3^). The initial concentration of lignohumate in solution was 0.1% wt. The solutions in vessels were stirred continuously (250 rpm). The decrease in the concentration of lignohumate in the solution was measured over time by means of UV/VIS spectrometer Hitachi U3900H (Hitachi, Tokyo, Japan). The concentration profiles in hydrogels were determined by the same method as in the case of the diffusion couple (described above). All experiments were performed at laboratory temperature (25 °C). Data are presented as average values with standard deviation bars.

## 4. Conclusions

In this study, the diffusion properties of lignohumate were studied in relation to its supramolecular character. The lignohumate, similarly to natural humic substances, contains particles with different sizes and shapes influencing their mobility and permeability into different materials. Since smaller particles are faster than bigger ones, their distribution in hydrogels changes with the distance from interface as well as with time of diffusion. The change in particle size distribution can result in the re-arrangement of secondary structure of lignohumate. The accumulation of bigger humic particles was observed in the donor hydrogel of diffusion couple, in spite of the fact that measured particle sizes were sufficiently small in comparison with pore sizes in the hydrogel. The re-arrangement of the lignohumate secondary structure as a result of the depletion of small particles can support the aggregation of lignohumate connected with the loss of its mobility. This phenomenon can affect the penetration of lignohumate into plant organs in its use for agricultural and horticultural purposes, its potential sedimentation, and in the decrease in active sites accessible for other substances in natural systems. In contrast, the destruction of weak bonds in the supramolecular structure of lignohumate can result in the formation of smaller particles with higher mobility. New bonds between particles can form and others are destroyed. The supramolecular structure changes are thus dependent on the amount and size distribution of lignohumate, its distance from the interface, and time. Our results show that the supramolecular structure of lignohumate can react sensitively on actual changes in its environs and thus affect their mobility and permeability into the hydrogel.

## Figures and Tables

**Figure 1 molecules-25-05831-f001:**
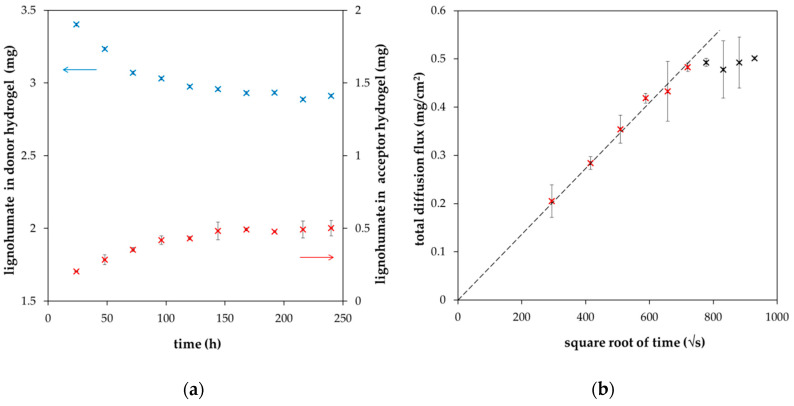
The time development of total amounts of lignohumate in donor (blue) and acceptor (red) hydrogels (**a**); experimental data fitted by Equation (2) (**b**).

**Figure 2 molecules-25-05831-f002:**
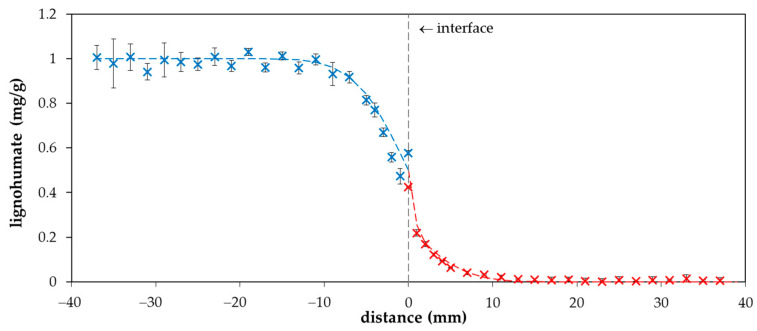
Concentration profile of lignohumate in agarose hydrogel after one day. The blue points belong to the donor hydrogel (distance *x* < 0), and the red points belong to the acceptor hydrogel (distance *x* > 0).

**Figure 3 molecules-25-05831-f003:**
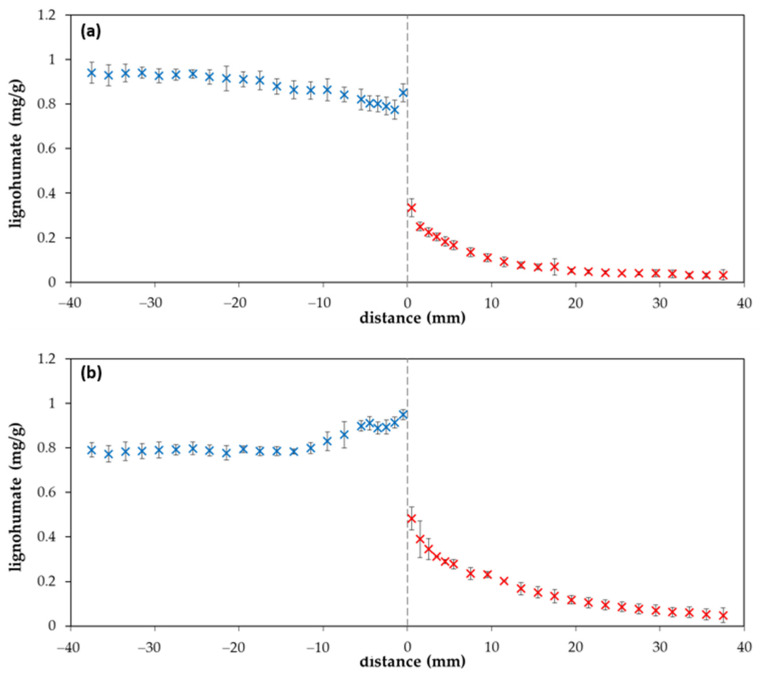
Concentration profiles of lignohumate in agarose hydrogel after two (**a**) and ten (**b**) days. The blue points belong to the donor hydrogel (distance *x* < 0), the red points belong to the acceptor hydrogel (distance *x* > 0).

**Figure 4 molecules-25-05831-f004:**
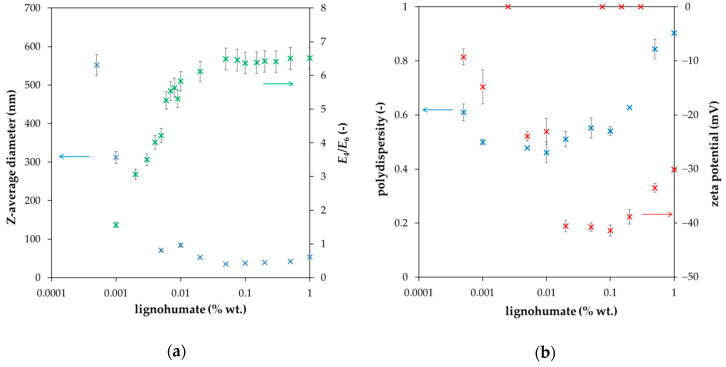
Z-average diameter and *E*_4_/*E*_6_ ratio (**a**); polydispersity and zeta potential of lignohumate in the dependence on its concentration in aqueous solution (**b**).

**Figure 5 molecules-25-05831-f005:**
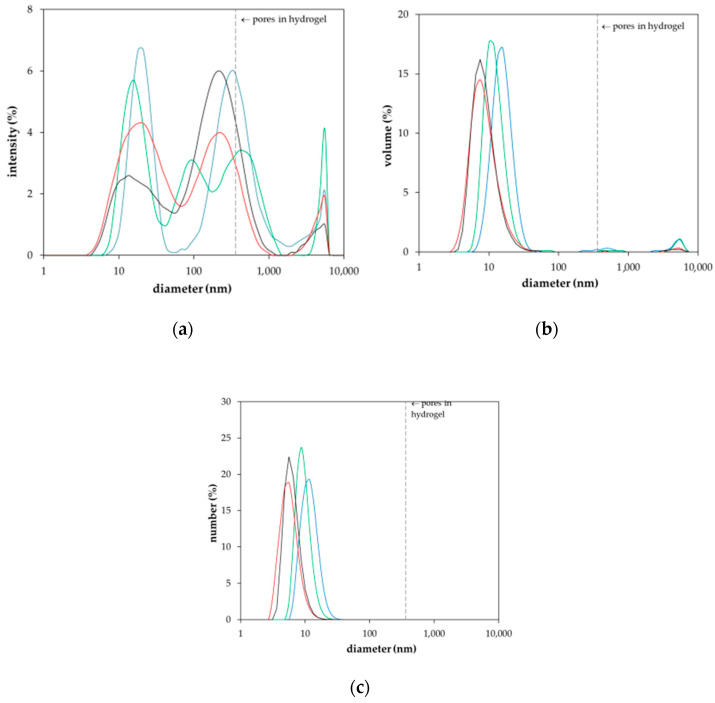
Intensity (**a**), volume (**b**) and number (**c**) particle size distributions of lignohumate with concentration 0.001% wt. (blue), 0.01% wt. (green), 0.1% wt. (red) and 1% wt. (black).

**Figure 6 molecules-25-05831-f006:**
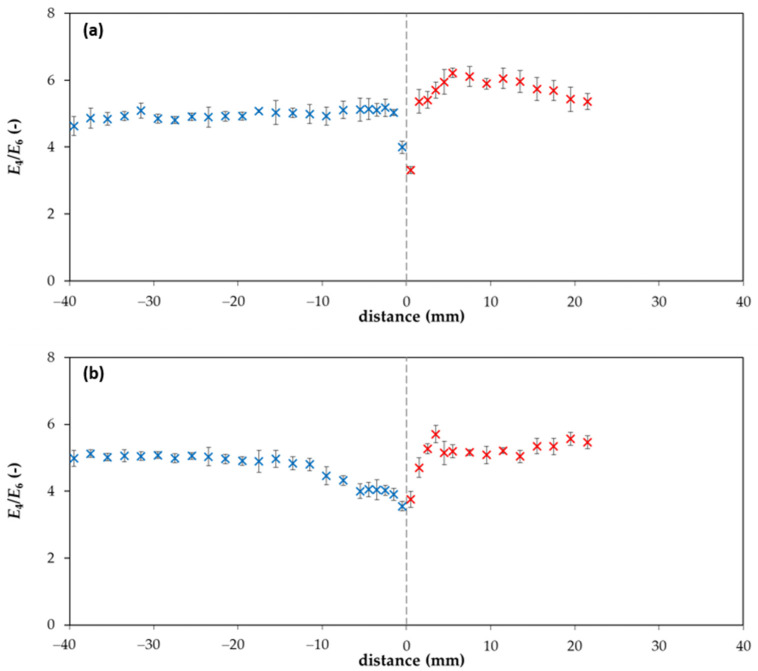
*E*_4_/*E*_6_ ratio in agarose hydrogel after two (**a**) and ten (**b**) days. The blue points belong to the donor hydrogel (distance *x* < 0), while the red points belong to the acceptor hydrogel (distance *x* > 0).

**Figure 7 molecules-25-05831-f007:**
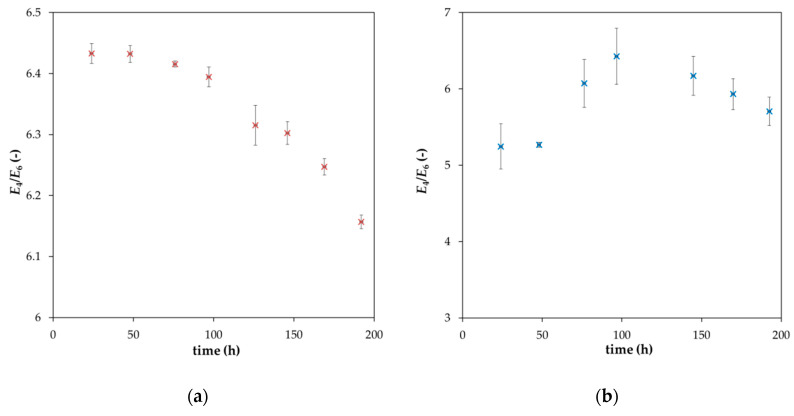
*E*_4_/*E*_6_ ratio of lignuhumate in its solution during its diffusion into the hydrogel (**a**); time development of *E*_4_/*E*_6_ ratio of lignohumate in the hydrogel at a distance of 0.5 mm from the interface (**b**).

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
