# Peer review of "How the Supramolecular Nature of Lignohumate Affects Its Diffusion in Agarose Hydrogel"

_molecules, 2020, doi:10.3390/molecules25245831_

Round 1

Reviewer 1 Report

Title: How Supramolecular Nature of LignohumateAffect Its Diffusion in Agarose Hydrogel

My Recommendation: Publish after minor revisions noted.

Comments:

The manuscript abounds fruitful discussion. I recommend publication of the manuscript after considering some minor remarks.

Which are the CAS Numbers of used chemicals in Experimental part?

They were used without further purification, or they are purified ? If chemicals were purified, how it was done ?

The English language, typos, grammatical errors throughout the Manuscript should be carefully revised.

The experiments seem to be carried out carefully and thus the data are reliable. The treatment of data is correct and the obtained results are new and interesting. This paper could be suitable for publication in this journal Molecules, while some improvements as mentioned in the letter above have been made.

Author Response

The manuscript abounds fruitful discussion. I recommend publication of the manuscript after considering some minor remarks.

Thank you for your positive review and recommendation.

Which are the CAS Numbers of used chemicals in Experimental part?

The CAS number of agarose is 9012-36-6 (added in the experimental part of manuscript). No CAS number was assigned to lignohumate.

They were used without further purification, or they are purified ? If chemicals were purified, how it was done ?

Agarose and lignohumate were used as purchased (without purification).

The English language, typos, grammatical errors throughout the Manuscript should be carefully revised.

The manuscript was revised by English professional. The language was improved.

 All changes are in the manuscript are in red.

Reviewer 2 Report

The manuscript is investigation of diffusion of lignohumate in agarose hydrogels.  The authors fully investigated diffusion coefficient of lignohumate in agarose hydrogels and considered the difference in the diffusion from the size due to the aggregation (supramolecular structure) of lignohumate.  I have one question.  As I wrote in 6, I am concerned about the difference in diffusion coefficient of lignohumate depending on the state.  After the authors have addressed the comments below, this work would be suitable for publication in Molecules.

  1. There is no Figure 2. The author should add Figure 2.
  2. 3 and 6 should be revised to make it easier for the reader to understand. What do blue and red mean?  Is the minus and minus of the x-axis the depth of the donor gel and acceptor gel, respectively? 
  3. Is there a difference in diffusion of lignofumate due to the difference in gel placement in the dispersion of lignohumate donor gel to acceptor gel? In other words, is there a difference in dispersity of lignohumate between vertically stacked and horizontally stacked?
  4. Does the dispersion rate of lignohumate increase as the gel contact area increases in the experiment of diffusion of lignohumate from donor gel to acceptor gel?
  5. Does the dispersion rate of lignohumate increase as the gel surface area increases in the experiment of diffusion of lignohumate from solution to hydrogel?
  6. How does the diffusion coefficient of lignohumate from donor gel to acceptor gel, in the gel, and from solution to gel compare to diffusion coefficient of lignohumate in solution? If not, what is the cause?

Author Response

1. There is no Figure 2. The author should add Figure 2.

I appologize, it is my misstake. Figure 2 was added in the manuscript.

2. 3 and 6 should be revised to make it easier for the reader to understand. What do blue and red mean?  Is the minus and minus of the x-axis the depth of the donor gel and acceptor gel, respectively? 

The blue points belong to the donor hydrogel (minus of the axis) and the red points belong to the acceptor hydrogel. The values on the x-axis represent the distance for the interface between both parts of diffusion couple. The explanation was added in figure captures.

3. Is there a difference in diffusion of lignofumate due to the difference in gel placement in the dispersion of lignohumate donor gel to acceptor gel? In other words, is there a difference in dispersity of lignohumate between vertically stacked and horizontally stacked?

The term “lignofumate” is my typing error. I appologize, only lignohumate is used in this study.

4. Does the dispersion rate of lignohumate increase as the gel contact area increases in the experiment of diffusion of lignohumate from donor gel to acceptor gel?

I am not sure if I understand correctly. What is the “dispersion rate”? Do you mean the “diffusion rate” (?). The diffusion rate depends on two parameters: the diffusion coefficient and the concentration gradient. The effective diffusion coefficient determined in this study should be the same for donor and acceptor hydrogel because the characters of both parts of diffusion couple do not differ and it seems that the value of diffusion coefficient does not depend on the lignohumate concentration. The diffusion rate is thus influenced mainly by concentration gradient which is the biggest one in the beginning of the experiment. In this study, the lignohumate changed its molecular organization during the diffusion and some particles accumulated at the interface between donor and acceptor hydrogels. Bigger particles (aggregates, aglomerates etc.) were decelerated and practically intercepted at the interface. In, contrast, the migration of small particles was supported and they diffused relatively quickly into the acceptor hydrogel. However, it cannot be declared in general, that the diffusion rate increased.

5. Does the dispersion rate of lignohumate increase as the gel surface area increases in the experiment of diffusion of lignohumate from solution to hydrogel?

The gel surface area did not increase. The hydrogel was placed in the same type of cuvette as in the case of diffusion couple. Therefore the area of hydrogel accessible for the diffusion was the same as the area of interface between donor and acceptor part in the diffusion couple.

6. How does the diffusion coefficient of lignohumate from donor gel to acceptor gel, in the gel, and from solution to gel compare to diffusion coefficient of lignohumate in solution? If not, what is the cause?

The diffusion coefficient is valid for the medium in which the diffusion is realized. It is not the diffusion “from... to...”. It is the diffusion in the hydrogel (with corresponding value of diffusion coefficient) and the diffusion in the solution (with other value of diffusion coefficient). The principal property of diffusion medium is their viscosity (see also equation 3). It means that the diffusion coefficient of the same particle can change at the interface between two different mediums (e.g. solution/hydrogel) and the diffusion coefficient in the hydrogel is generally the same without regard to the character of donor medium. In our study, the diffusion experiments from the solution to the hydrogel were realized in order to understand better the observed accumulation of lignohumate at the interface. If the diffusion is realized in the couple of two hydrogels, the lignohumate has a concentration gradient in both parts (donor and acceptor). In contrast, the solution can be stirred, therefore it has no concentration gradient in the donor solution. In addition, the particle size distribution can be determined directly in the solution (not only indirectly by means of E4/E6 ratio).

 All changes in the manuscript are in red.